# Taxonomy and Phylogeny of Corticioid Fungi in Auriculariaceae (Auriculariales, Basidiomycota): A New Genus, Five New Species and Four New Combinations

**DOI:** 10.3390/jof9030318

**Published:** 2023-03-03

**Authors:** Yue Li, Ting Nie, Karen K. Nakasone, Hai-Jiao Li, Shuang-Hui He

**Affiliations:** 1School of Ecology and Nature Conservation, Beijing Forestry University, Beijing 100083, China; 2Qingyuan Forestry Bureau, Qingyuan 511500, China; 3Center for Forest Mycology Research, Northern Research Station, U.S. Forest Service, Madison, WI 53726, USA; 4National Institute of Occupational Health and Poison Control, Chinese Center for Disease Control and Prevention, Beijing 100050, China

**Keywords:** Agaricomycetes, *Eichleriella*, Heterobasidiomycetes, *Heterocorticium*, *Heteroradulum*

## Abstract

The Auriculariaceae accounts for most of the species in the Auriculariales, and all species in the family are wood-decaying fungi with gelatinous, crustaceous, or woody basidiomes. Many new taxa were published recently, but the taxonomy and phylogeny of the corticioid species in the Auriculariaceae are far from resolved. We undertook a comprehensive taxonomic and phylogenetic study of the family with emphasis on corticioid specimens collected from East and Southeast Asia. Phylogenetic analyses on concatenated ITS and 28S rDNA sequences of representative taxa of the Auriculariaceae and the genera *Eichleriella* and *Heteroradulum* were carried out that resolved five new lineages. *Heterocorticium* gen. nov. is established for two species with resupinate coriaceous basidiomes with smooth, pigmented hymenophores. Five new species, *H. bambusicola* (generic type), *H. latisporum*, *Eichleriella alpina*, *E. bambusicola,* and *Heteroradulum maolanense*, are described and illustrated. In addition, *Heterochaete delicata*, *H. discolor,* and *H. sinensis* are transferred to *Eichleriella*, whereas *H. roseola* is regarded as a synonym of *Kneiffia discolor (= H. discolor)*. *Eichleriella aculeobasidiata* is treated as a synonym of *Heterochaete sinensis* (= *E. sinensis*). *Heterochaete mussooriensis* is transferred to *Heteroradulum* with *Heteroradulum semis* as a heterotypic synonym. The present study contributes to the understanding of species diversity, taxonomy, and phylogeny of corticioid fungi in Asia.

## 1. Introduction

Auriculariaceae is the largest and best-supported clade in the Auriculariales (Basidiomycota) and consists of a large group of wood-decaying fungi with varied basidiome configurations [1,2,3,4] (Figure 1). Within the family are the well-known jelly fungi in *Auricularia* Bull. and *Exidia* Fr. that have been intensively studied by several authors in recent years, although the latter genus remains polyphyletic [5,6,7,8,9,10,11]. Species with poroid hymenophores in *Aporpium* Bondartsev & Singer, *Elmerina* Bres., and *Protodaedalea* Imazeki are also members of the Auriculariaceae [12,13,14]. Corticioid and stereoid taxa are numerous in Auriculariaceae and typically classified in three main genera, *Eichleriella* Bres., *Exidiopsis* (Bref.) A. Møller, and *Heterochaete* Pat. [15,16,17,18,19]. The genera were distinguished primarily by the morphology of the basidiome and hymenophore. Our understanding of these genera has changed dramatically since the phylogenic study of Malysheva and Spirin [15]. They showed that the generic types of *Heterochaete* and *Exidiopsis*, *H. andina* Pat. & Lagerh. and *E. effusa* (Bref. ex Sacc.) A. Møller, respectively, were placed together in a well-supported lineage, and other species of *Heterochaete* were nested within the *Eichleriella* s.s. and *Heteroradulum* Lloyd ex Spirin & Malysheva clades. They reintroduced *Heteroradulum* to accommodate *Eichleriella kmetii* Bres. and related species that are distantly related to *Eichleriella* s.s. [15,20]. In addition, two new genera, *Amphistereum* Spirin & Malysheva and *Sclerotrema* Spirin & Malysheva, were erected [15]. Subsequently, several phylogenetic studies revealed additional new lineages, viz. *Adustochaete* Alvarenga & K.H. Larss., *Alloexidiopsis* L.W. Zhou & S.L. Liu, *Crystallodon* Alvarenga, *Metulochaete* Alvarenga, and *Proterochaete* Spirin & Malysheva [4,21,22,23].

Microscopically, taxa in the Auriculariaceae and in the Auriculariales share many common morphological characters [3]. Except for *Auricularia* that has transversely septate basidia, all other species have typical tremellaceous basidia that are longitudinally septate. Dikaryophyses are usually present in enclosing the basidia, and basidiospores are mostly cylindrical to broadly ellipsoid and relatively large.

Many new genera and species have been described recently; nevertheless, the species diversity of the corticioid fungi in the Auriculariaceae in subtropical and tropical Asia has not been sufficiently explored. The taxonomy and phylogenetic relationships of previously described taxa from the region need to be studied and integrated with the newer taxa. In the present study, we performed (1) phylogenetic analyses of Auriculariaceae by adding corticioid taxa recently collected from East and Southeast Asia and (2) morphological examinations of recent collections as well as type specimens of some older species. We resolved five new lineages, including a new genus named *Heterocorticium.* Five new species in *Heterocorticium*, *Eichleriella,* and *Heteroradulum* were described and illustrated. In addition, the taxonomic and phylogenetic positions of some species of *Heterochaete* and *Eichleriella* were determined, resulting in the proposal of several new combinations.

## 2. Materials and Methods

Specimen collection, morphological studies, DNA extraction and sequencing and phylogenetic analyses followed [24]. Three separate datasets of concatenated ITS-28S sequences of the Auriculariaceae, *Eichleriella* and *Heteroradulum* were analyzed (Table 1). *Protoacia delicata* Spirin & Malysheva was selected as the outgroup for the Auriculariaceae dataset, *Amphistereum leveilleanum* (Berk. & M.A. Curtis) Spirin & Malysheva and *A. schrenkii* (Burt) Spirin & Malysheva were used for the *Eichleriella* dataset, whilst *Exidiopsis effusa* (Bref. ex Sacc.) Möller and *Tremellochaete japonica* (Yasuda) Raitv. were used for the *Heteroradulum* dataset. Four Markov chains were run for 1,000,000 generations for *Eichleriella* and *Heteroradulum* datasets separately, and 4,000,000 generations for the Auriculariaceae dataset until the split deviation frequency value was lower than 0.01.

## 3. Results

### 3.1. Phylogenetic Analyses

The Auriculariaceae dataset contained 58 ITS and 57 28S sequences from 61 samples representing 47 ingroup taxa and the outgroup (Table 1), and had an aligned length of 1953 characters, of which 414 were parsimony-informative. MP analysis yielded 33 equally parsimonious trees (TL = 1864, CI = 0.436, RI = 0.657, RC = 0.287, HI = 0.564). The *Eichleriella* dataset contained 30 ITS and 26 28S sequences from 31 samples representing 17 ingroup taxa and the outgroup and had an aligned length of 1859 characters, of which 137 were parsimony-informative. MP analysis yielded six equally parsimonious trees (TL = 409, CI = 0.650, RI = 0.793, RC = 0.516, HI = 0.350). The *Heteroradulum* dataset contained 20 ITS and 20 28S sequences from 20 samples, representing nine ingroup taxa and the outgroup, and had an aligned length of 1847 characters, of which 115 were parsimony-informative. MP analysis yielded one equally parsimonious tree (TL = 281, CI = 0.726, RI = 0.835, RC = 0.607, HI = 0.274).

jModelTest suggested SYM+I+G as the best-fit models of nucleotide evolution for the Auriculariaceae dataset, and GTR+I+G for the *Eichleriella* and *Heteroradulum* datasets. The average standard deviations of split frequencies of BI were 0.008874, 0.007964, and 0.003569 for the three datasets at the end of the runs. The MP, ML, and BI analyses of the three datasets resulted in almost identical tree topologies. The MP tree of the Auriculariaceae is shown in Figure 2, whilst ML trees of *Eichleriella* and *Heteroradulum* are shown in Figure 3 and Figure 4 with the parsimony bootstrap values (≥ 50%, first), likelihood bootstrap values (≥ 50%, second), and Bayesian posterior probabilities (≥ 0.95, third) labelled along the branches.

In the Auriculariaceae tree (Figure 2), 19 distinct lineages corresponding to 18 known genera and the new genus, *Heterocorticium*, were recognized. Two new species of *Heterocorticium*, *H. bambusicola* and *H. latisporum*, formed a distinct lineage with strong support values (93/94/1). The lineages representing *Eichleriella* and *Heteroradulum* were strongly supported as monophyletic genera. In the *Eichleriella* tree (Figure 3), seventeen distinct lineages including two new species, *E. alpina* and *E. bambusicola*, and clades representing three *Heterochaete* species, *H. delicata*, *H. discolor*, *H. sinensis*, were recognized. Two samples of *E. aculeobasidiata,* including the holotype (CLZhao 6159) and one sample identified as *Heterochaete delicata* (TUFC33717), were nested within the *E. sinensis* lineage (61/73/1). In the *Heteroradulum* tree (Figure 4), eight lineages including the species, *H. maolanense* and *H. mussooriense*, were recognized. The holotype of *H. semis* (OM10618) was nested within the *H. mussooriense* lineage (99/93/1).

### 3.2. Taxonomy

***Heterocorticium*** S.H. He, T. Nie & Yue Li, **gen. nov.**

MycoBank: MB847441

Type species—*Heterocorticium bambusicola* S.H. He, T. Nie & Yue Li

Etymology—“*Hetero*-”: different, refers to the septate basidia; “*Corticium*”: a generic name of corticioid homobasidiomycetes, refers to the basidiomes similar to *Corticium*.

Basidiomes annual, resupinate, effused, closely adnate, inseparable from substrate, thin, coriaceous. Hymenophore smooth, grey, orange to brown; margin thinning out or abrupt, adnate. Hyphal system monomitic; generative hyphae clamped, colorless, thin- to thick-walled, frequently branched, septate. Subiculum indistinct, composed of densely interwoven hyphae. Cystidia absent or present. Dikaryophyses present, colorless, thin-walled, frequently branched. Basidia ovoid or subglobose, longitudinally septate, four-celled, embedded, without enucleate stalk. Basidiospores cylindrical or ellipsoid with an apiculus, colorless, thin-walled, smooth, IKI–, CB–, with oily contents, capable of germinating by repetition.

Notes—*Heterocorticium* is characterized by the resupinate coriaceous basidiomes with smooth, pigmented hymenophores, monomitic hyphal system with clamped generative hyphae, and cylindrical or ellipsoid basidiospores. Macroscopically, *Heterocorticium* resembles a typical crust, but can be readily distinguished from homobasidiomycetes by its longitudinally septate basidia. *Heterocorticium* is similar to *Exidiopsis* and *Alloexidiopsis* in that they differ by pale-colored basidiomes [4,17,18]. The monotypic genus *Sclerotrema* recently segregated from *Exidiopsis* is similar to *Heterocorticium* by sharing a pigmented hymenophore but differs by its distinctly curved allantoid basidiospores and occurrence mainly on *Alnus* [15,18]. In the phylogenetic tree, *Heterocorticium* formed a distinct lineage and did not show a close relationship with other genera (Figure 2).

***Heterocorticium bambusicola*** S.H. He, T. Nie & Yue Li, **sp. nov.** Figure 5A, Figure 6 and Figure 7

MycoBank: MB847442

Type—China, Jiangxi Province, Yifeng County, Guanshan Nature Reserve, on culm of dead bamboo, 10 August 2016, He 4236 (BJFC 023678, holotype, CFMR, isotype).

Etymology—refers to its preferred substrate, bamboo.

Fruiting body—Basidiomes annual, resupinate, effused, closely adnate, inseparable from substrate, coriaceous, first as small colonies, later confluent up to 20 × 5 cm, up to 120 µm thick in section. Hymenophore smooth, orange white (6A2), orang grey (6B2), greyish orange [6B(3–6)] to brownish orange [6C(3–6)], not cracked; margin thinning out or abrupt, distinct and white (6A1) when juvenile, indistinct and concolorous with hymenophore when mature.

Microscopic structures—Hyphal system monomitic; generative hyphae with clamp connections. Subiculum thin, with many crystals adjacent to substrate; hyphae in this layer colorless, thin-walled, frequently branched and septate, slightly agglutinated, densely interwoven, 1–3 µm in diam. Cystidia absent. Dikaryophyses numerous, forming a light brown layer in hymenium, colorless, thin-walled, frequently branched. Basidia embedded, ovoid to subglobose, longitudinally septate, four-celled, without enucleate stalk, 13–20 × 9–11 µm. Basidiospores cylindrical with a distinct apiculus, slightly curved, colorless, thin-walled, smooth, IKI–, CB–, with oily contents, capable of germinating by repetition, 10–14 × 5–6 µm, L = 11.5 µm, W = 5.5 µm, Q = 1.8–2.4 (*n* = 60/2).

Additional specimens examined—China, Fujian Province, Jian’ou County, Wanmulin Nature Reserve, on culm of dead bamboo, 19 August 2016, He 4545 (BJFC 023986); Nanping County, Wuyishan Forest Park, on dead bamboo, 3 October 2018, He 5691 (BJFC 026753); Guangxi Autonomous Region, Xing’an County, Mao’ershan Nature Reserve, on culm of dead bamboo, 13 July 2017, He 4799 (BJFC 024318, CFMR); Jiangxi Province, Yifeng County, Guanshan Nature Reserve, on culm of dead bamboo, 10 August 2016, He 4234 (BJFC023676) and He 4260 (BJFC 023702); Anyuan County, Sanbaishan Forest Park, on culm of dead bamboo, 15 August 2016, He 4415 (BJFC 023856); Taiwan, Nantou County, Jenai Township, Entrance of Southern Tungyenshan, alt. 1700 m, on culm of dead bamboo, 7 December 2016, He 4604 (BJFC 024046, CFMR) and He 4607 (BJFC 024049); Xitou, on culm of rotten bamboo, 11 December 2016, He 4638 (BJFC 024081); Yunnan Province, Lushui County, Gaoligongshan Nature Reserve, on culm of dead bamboo, 29 November 2015, He 3341 (BJFC 021736, CFMR), He 3344 (BJFC 021739) and He 3359 (BJFC 021754); Xichou County, Xiaoqiaogou Forest Farm, on culm of dead bamboo, 16 November 2019, He 6293 (BJFC 033237); Malaysia, Kuala Lumpur, Forest Eco-Park, on culm of dead bamboo, 8 December 2019, He 6430 (BJFC 033374) and He 6435 (BJFC 033379). 

Notes—*Heterocorticium bambusicola* is widely distributed in southern China on bamboo and was also found in Malaysia. It is characterized by thin, resupinate, coriaceous basidiomes, smooth hymenophore, and well-developed dikaryophyses. The other species in the genus, *H. latisporum*, differs from *H. bambusicola* by having grayish brown hymenophore and wider basidiospores.

***Heterocorticium latisporum*** S.H. He, T. Nie & Yue Li, **sp. nov.** Figure 5B and Figure 8

MycoBank: MB847443

Type—China, Sichuan Province, Qionglai County, Tiantaishan Forest Park, on fallen angiosperm branch, 23 September 2012, He 20120923-20 (BJFC 014667, holotype, CFMR, isotype).

Etymology—refers to wide basidiospores.

Fruiting body—Basidiomes annual, resupinate, effused, closely adnate, inseparable from substrate, coriaceous; first as small, scattered colonies, later confluent up to 10 × 2.5 cm, up to 150 µm thick in section. Hymenophore smooth, brownish orange [6C(3–4)], greyish brown [6(D–F)3], light brown [6D(4–6)] to brown [6E(4–6)], not cracked or locally cracked; margin thinning out, white (6A1) when juvenile, becoming concolorous with hymenophore when mature.

Microscopic structures—Hyphal system monomitic; generative hyphae with clamp connections. Subiculum thin, with scattered crystals; hyphae colorless, thin- to slightly thick-walled, frequently branched, septate, slightly agglutinated, densely interwoven, 2–3 µm in diam. Cystidia clavate, colorless, with a basal clamp connection, 20–60 × 6–13 µm. Dikaryophyses numerous, colorless, thin-walled, slightly branched. Basidia embedded, ovoid to subglobose, longitudinally septate, four-celled, without enucleate stalk, 16–22 × 11–15 µm. Basidiospores broadly ellipsoid-to-ovoid with a distinct apiculus, colorless, thin-walled, smooth, IKI–, CB–, with oily contents, capable of germinating by repetition, 11–13 (–14) × 8–9 µm, L = 12 µm, W = 8.5 µm, Q = 1.4 (*n* = 30/1).

Additional specimen examined—China, Anhui Province, Jingde County, Majiaxi Forest Park, on dead angiosperm branch, 27 July 2022, He 7540 (BJFC 038676).

Notes—*Heterocorticium latisporum* is characterized by resupinate coriaceous basidiomes with a grayish brown, smooth hymenophore, and broadly ellipsoid-to-ovoid basidiospores.

**Eichleriella alpina** S.H. He, T. Nie & Yue Li, **sp. nov.** Figure 5C and Figure 9

MycoBank: MB847444

Type—China, Sichuan Province, Xiaojin County, Siguniangshan Nature Reserve, on dead angiosperm branch, 16 September 2012, He 20120916-1 (BJFC 014581, holotype).

Etymology—refers to growing on mountains of high altitude.

Fruiting body—Basidiomes annual, discoid, resupinate to slightly effused-reflexed, adnate, separable from substrate, coriaceous, brittle; first as small patches, later confluent up to 5.5 cm long, 1.5 cm wide, up to 400 µm thick in section. Hymenophore smooth to slightly tuberculate, white (6A1) when fresh, becoming greyish green [1D(2–5)], yellowish grey (2C2) to olive grey (2D2) upon drying, not cracked or partly cracked when dry; margin abrupt, slightly elevated, concolorous or slightly darker than hymenophore.

Microscopic structures—Hyphal system monomitic, generative hyphae with clamp connections. Basal layer present, yellow to yellowish brown; hyphae yellow, distinctly thick-walled, densely agglutinated, unbranched, aseptate, 3–5 µm in diam. Subiculum thick; hyphae in this layer colorless, slightly-to-distinctly thick-walled, rarely branched, moderately septate, interwoven, 2–4 µm in diam. Hymenium composed of dikaryophyses, basidia and immature basidia. Cystidia absent. Dikaryophyses numerous, colorless, thin-walled, frequently branched. Basidia clavate to subcylindrical, colorless, longitudinally septate, four-celled, 20–26 × 10–15 µm. Immature basidia abundant, subglobose to ellipsoid. Basidiospores cylindrical with a small apiculus, slightly curved, colorless, thin-walled, smooth, IKI–, CB–, with oily contents, capable of germinating by repetition, 15–17 × 6–7 µm, L = 16 µm, W = 6.5 µm, Q = 2.5 (*n* = 30/1).

Notes—*Eichleriella alpina* is characterized by having discoid-to-slightly effused-reflexed basidiomes, a distinct basal layer, relatively large basidiospores and a distribution in temperate areas of southwestern China. *Eichleriella macrospora* (Ellis and Everh.) G.W. Martin is similar but differs in having slightly shorter basidiospores (10–15 × 5–7 µm) and a distribution in the north central USA [33]. In the phylogenetic tree, *E. alpina* formed a distinct lineage in the *Eichlerilla* clade (Figure 3).

**Eichleriella bambusicola** S.H. He, T. Nie & Yue Li, **sp. nov.** Figure 5D and Figure 10

MycoBank: MB847445

Type—Thailand, Chiang Rai Province, Doi Mae Salong, on culm of dead bamboo, 22 July 2016, He 4073 (BJFC 023514, holotype, CFMR, isotype).

Etymology—refers to growing on bamboo.

Fruiting body—Basidiomes annual, resupinate, adnate, separable from substrate, soft, membranous to coriaceous, first as small colonies, later confluent up to 10 cm long, 4 cm wide, up to 500 µm thick in section, new basidiomes are usually grown from dead parts. Hymenophore smooth, greyish orange [5B(3–5)] to brownish orange [5C(3–5)] in fertile parts, light brown [5D(3–5)] in sterile or dead areas, not cracked; margin slightly thinning out, white (5A1), fimbriate when juvenile, becoming concolorous or darker than hymenophore with age.

Microscopic structures—Hyphal system dimitic, generative hyphae with clamp connections. Subiculum thick, yellow. Skeletal hyphae setae-like, predominant, yellow to yellowish brown, distinctly thick-walled to subsolid, frequently branched, usually bifurcated, up to 7 µm in diam., slightly dextrinoid, branches short or long with acute tips. Generative hyphae rare to abundant, colorless, thin- to slightly thick-walled, rarely branched, frequently septate, scattered among skeletal hyphae, 1.5–4 µm in diam. Hymenium composed of dikaryophyses, basidia and immature basidia. Cystidia absent. Dikaryophyses numerous, colorless, thin-walled, frequently branched, usually stalked, swollen in middle part. Basidia ovoid to subglobose, colorless, longitudinally septate, four-celled, 15–20 × 9–13 µm. Basidiospores ellipsoid-to-broadly ellipsoid, colorless, thin-walled, smooth, IKI–, CB–, with oily contents, capable of germinating by repetition, 10–13 × 6–7.5 µm, L = 11 µm, W = 7 µm, Q = 1.6–1.7 (*n* = 60/2).

Additional specimens examined—China, Hunan Province, Dong’an County, Shunhuangshan Nature Reserve, on culm of dead bamoo, 3 July 2015, He 2365 (BJFC 020819, CFMR) and He 2371 (BJFC 020825, CFMR); Jiangxi Province, Yifeng County, Guanshan Nature Reserve, on culm of dead bamboo, 9 August 2016, He 4168 (BJFC 023610, CFMR); Zhejiang Province, Lin’an County, Tianmushan Nature Reserve, on culm of dead *Phyllostachys*, 15 October 2004, Dai 6391 (BJFC 016628, CFMR); Thailand, Chiang Rai Province, Doi Pui, on culm of dead bamboo, 23 July 2016, He 4088 (BJFC 023529).

Notes—*Eichleriella bambusicola* is characterized by having resupinate basidiomes, a dimitic hyphal system with yellow setae-like skeletal hyphae, ellipsoid-to-broadly ellipsoid basidiospores, and fruiting on bamboo in subtropical and tropical areas of China. *Eichleriella bambusicola* is similar to *E. leveilleana* (Berk. and M.A. Curtis) Burt and *E. schrenkii* Burt because of its dimitic hyphal system with richly dichotomously branched and yellow-to-brown skeletal hyphae. This character was mentioned by Malysheva and Spirin [15] when establishing the genus *Amphistereum* for the latter two species. Notably, *E. bambusicola* nested within the *Eichleriella* s.s. lineage rather than the *Amphistereum* lineage in our phylogenetic analyses, although the two lineages are sisters with relatively strong support values in MP and ML analyses (Figure 2). *Eichleriella tenuicula* (Lév.) Spirin and Malysheva also has a dimitic hyphal system but differs from *E. bambusicola* by having a spiny hymenophore, unbranched skeletal hyphae, and two-celled basidia [15,16]. In the phylogenetic tree, samples of *E. bambusicola* from Thailand and China formed a distinct linage (Figure 3). The yellow, setae-like skeletal hyphae of *E. bambusicola* resemble the dichohyphae of *Vararia* P. Karst, which is a homobasidiomycetes in the Russulales [34].

**Eichleriella delicata** (Bres.) S.H. He & Nakasone, **comb. nov.** Figure 11A

MycoBank: MB847446

≡ *Heterochaete delicata* Bres., Hedwigia 53: 77, 1912.

≡ *Hydnum delicatum* Klotzch ex Berk., Ann. Nat. Hist., Mag. Zool. Bot. Geol. 3 no. 19: 395, 1839 (nom. illeg.); non *H. delicatum* Schwein., Trans. Am. Phil. Soc., New Series 4(2): 161. 1832.

Specimens studied—*Eichleriella delicata*: Australia, New South Wales, Sydney, Hyde Park, on hardwood, 15 October 2001, H.H. Burdsall, Jr., FP-140099 (CFMR); China, Yunnan Province, Ruili County, Moli Tropical Rain Forest Park, on fallen angiosperm branch, 2 December 2015, He 3469 (BJFC 021866, CFMR).

*Eichleriella tenuicula*: China, Hainan Province, Danzhou Tropical Botanical Garden, on fallen angiosperm branch, 7 May 2009, Cui 6306 (BJFC 004162); Taiwan, Nantou County, Xinyi Township, Xitou, on fallen angiosperm branch, 11 December 2016, He 4634 (BJFC 024077); Yunnan Province, Ruili County, Moli Tropical Rain Forest Park, on fallen angiosperm branch, 2 December 2015, He 3483 (BJFC 021880, CFMR); Thailand, Chiang Rai Province, Campus of Mae Fah Luang University, on dead but still attached branch of *Cinnamomum porrectum*, 21 July 2016, He 4053 (BJFC 023492).

Notes—Bodman [16] treated *Heterochaete tenuicula* (Lév.) Pat. as a synonym of *H. delicata*, whereas Roberts [35] preferred *H. tenuicula* and treated *H. delicata* as a synonym. Malysheva and Spirin [15] transferred *H. tenuicula* to *Eichleriella* but did not mention *H. delicata*. *Heterochaete delicata* was described from India whereas *H. tenuicula* was from Java, Indonesia. Morphologically, the two species are similar sharing effused-to-effuse-reflexed basidiomes, two-celled basidia, and long basidiospores [16,35]. However, the taxa are placed in distinct lineages in the phylogenetic tree (Figure 3). Although inseparable in morphology, we accept *H. delicata* and *H. tenuicula* as distinct species and propose the transfer of *H. delicata* to *Eichleriella*. Significantly, the ITS sequence similarity between *H. delicata* (He 3469) and *H. tenuicula* (He 3483) is 97.5 % of 514 base pairs.

**Eichleriella discolor** (Berk. & Broome) S.H. He & Nakasone, **comb. nov.** Figure 11B

MycoBank: MB847447

≡ *Kneiffia discolor* Berk. & Broome, Journal Linnean Society. Botany 14: 62, 1873 (1875).

≡ *Kneiffiella discolor* (Berk. & Broome) Henn., in Engler & Prantl, Nat. Pflanzenfam.: I, Teil. 1 (Leipzig), Abt.: Fungi (Eumycetes): 141, 1898.

≡ *Heterochaete discolor* (Berk. & Broome) Petch, Annals Royal Botanic Gardens Peradeniya 9: 137, 1924.

= *Heterochaete tonkiniana* Pat., Bulletin Herbier Boissier 1: 301, 1893.

= *Heterochaete roseola* Pat., Bulletin Société Mycologique France 29: 207, 1913.

= *Heterochaete cheesmanii* Wakef., Bulletin Miscellaneous Information, Kew 1915: 373.

Fruiting body—Basidiomes widely effused, closely adnate; first as small circular-to-orbicular colonies, later confluent, up to 7 cm long, 4 cm wide, thin, 100–300 (–900) μm thick, soft, subceraceous to cretaceous. Hymenophore spiny-to-velutinous from hyphal pegs, (4–) 6–8 (–10) per mm, with smooth areas between pegs; pegs above hymenium (70–) 85–200 (–240) × (30–) 45–60 (–80) μm, single, terete, brittle, colorless to concolorous first then becoming dark yellow or light brown; overall surface first yellowish white (4A2), pale yellow (4A3), orange white (5A2), pale orange (5A3), light orange (5A4), then darkening to greyish orange (5B3), brownish orange (5C5), yellowish brown (5D4); margins variable sometimes within a single collection, adnate to somewhat detached, distinct, abrupt, thick, raised, felty-silky, brownish yellow, usually darker than hymenophore, or appressed, adnate, thinning out with edges white, fimbriate, silky to woolly.

Microscopic structures—Hyphal system monomitic but appearing dimitic from sclerified hyphae in subiculum and pegs. Hyphal pegs arising deep in subiculum as a compact, agglutinated fascicle composed of an inner core of brown pigmented, sclerified subicular hyphae, then above the hymenium hyphae differentiating into dikaryophyses along sides and at apices, often encrusted, sometimes terminal dikaryophyses colorless and thin-walled throughout or yellow to brownish yellow and walls thickening toward base. Subiculum up to 150 (–700) μm thick, a partially agglutinated tissue composed of a dense, dark yellow-to-brown tissue of sclerified hyphae arranged parallel to substrate, giving rise to hyphal pegs, hyphae 2–4 μm diam, clamps inconspicuous or degraded, sparsely branched, walls yellow to brownish yellow, thin to 1 μm thick, smooth. Subhymenium up to 80 μm thick, a moderately dense, colorless tissue of loosely interwoven hyphae turning into hymenium, composed of hyphae, 2.5–3.5 μm in diameter, clamped, walls thin, colorless, smooth. Hymenium 25–100 μm thick, a dense palisade of dikaryophyses enclosing cystidia and basidia. Dikaryophyses abundant, 35–60 × 3.5–5 μm, clamped at base, simple or with short nodulose or knobby branches, walls colorless, thin, smooth. Cystidia scattered to numerous, embedded to barely projecting, cylindrical, clavate, or subfusiform, often with a stalk, apices rounded, subacute or bulbous, 40–70 × 7–13 μm, clamped at base, walls colorless, thin, smooth, with homogenous contents. Basidia longitudinally septate, four-celled, embedded with only tips of sterigmata observed beyond hymenium, ellipsoid to ovoid, (11–) 13–23 × 8.5–11 μm, with a small, basal clamp, 4-sterigmate, rarely with 2 or 3 sterigmata, walls colorless, thin, smooth. Basidiospores cylindrical, often ventrally depressed, (10–) 11–13.5 (–15) × 5–6 μm, L = 12.5 μm, W= 5.5 μm, Q = 2.3–2.4 (60/2), colorless, thin-walled, smooth, IKI–, CB–, germination sometimes observed.

Distribution—Australia, Cambodia, China, Nepal, New Zealand, Philippines, Sri Lanka, Thailand, Vietnam.

Type specimens examined—Australia, New South Wales, Moruya, on bare wood, W.N. Cheesman, 1914 (K(M)4481, holotype of *H. cheesmanii*); Sri Lanka, Central Province, on wood December 1868, G.H.K.Thwaites 982 (K(M)132234, holotype of *Kneiffia discolor*); Vietnam, Hanoi, on bark, 22 June 1911, Dupont no. 676 (FH–HUH00940160, holotype of *H. roseola*); Vietnam, Tonkin, Thanh Hoa Ngoc Au (Than Hoa), on bark and wood, 29 January 1892, H.F. Bon 5062 (FH-HUH01093670, holotype of *H. tonkiniana*).

Representative specimens examined—Australia, Victoria, Gippsland Highland, Tarra Valley, on fallen branches of orangewood, 12 Sept 1955, K. Healy, N.W.M. Walters, & E. DaCosta (MEL-2313649, as *H. discolor*); Cambodia, Reserv foretrie de Campong Chhnang, on (bark of) dead branches, July 1921, P.A. Pételot 360 (FH-HUH00940161; BPI-719711, as *H. roseola*); China, Gansu Province, Tianshui County, Maijishan Forest Park, on fallen angiosperm branch, 8 August 2015, He 2488 (BJFC 020941); Guangxi Autonomous Region, Tianlin County, Cenwanglaoshan Nature Reserve, on dead angiosperm branch, 8 July 2017, He 4708 (BJFC 024227, CFMR); Huanjiang County, Mulun Nature Reserve, on rotten angiosperm trunk, 10 July 2017, He 4763 (BJFC 024281); Guizhou Province, Libo County, Maolan Nature Reserve, on dead angiosperm branch, 14 June 2016, He 3771 (BJFC 022270); 11 July 2017, He 4778 (BJFC 024295); Hainan Province, Wanning County, Xinglong Botanical Garden, on fallen angiosperm trunk, 19 March 2016, He 3610 (BJFC 022108); Wuzhishan County, Wuzhishan Nature Reserve, on dead angiosperm branch, 10 June 2016, He 3947 (BJFC 022449); Hunan Province, Zhangjiajie County, Zhangjiajie Nature Reserve, on fallen angiosperm tree, 7 July 2015, He 2297 (BJFC 020752); Yizhang County, Mangshan Nature Reserve, on dead angiosperm branch, 26 July 2016, He 4008a (BJFC 023446); Taiwan, Nantou County, Jenai Township, Entrance of Southern Tungyenshan, alt. 1700 m, on fallen angiosperm trunk, 7 December 2016, He 4584 (BJFC 024026); Xizang Autonomous Region, Linzhi County, Lulang, on dead angiosperm trunk, 17 September 2010, He 321 (BJFC 021980, CFMR); on dead Hippophae branch, 17 September 2010, He 331 (BJFC 021982); Yunnan Province, Jingdong County, Ailaoshan Nature Reserve, on dead angiosperm branch, 25 August 2015, He 2620 (BJFC 021066, CFMR); Baoshan County, Gaoligongshan Nature Reserve, Baihualing, on dead Quercus branch, He 3401 (BJFC 021797); Nepal, Chitwan National Park, Narayani River, Island Jungle Resort, on hardwood twig, 15 March 2008, T. Rödel, NP/13 (Rödel private herbarium, CFMR); New Zealand, Weraroa, Wellington, on fallen log, 30 October 1919, G.H. Cunningham 353 (BPI-719737, as *H. tenuicula*); Philippines, Luzon, Mt. Maquiling, on dead wood, 28 Oct 1920, F. Obias, ex Herb. O.A. Reinking no. 10516 (BPI-719764, FH-HUH00940152, as *H. leveillei*); Thaliand, Chiang Rai Province, Doi Mae Salong, on dead angiosperm branch, 22 July 2016, He4069 (BJFC 023510); Chiang Mai Province, Doi Saket, on dead angiosperm branch, 24 July 2016, He 4107 (BJFC 023549); Vietnam, Thac Mai Preservation Park, lowland rain forest Dipterocarp, on dead angiosperm branch, 14 October 2017, He 5211 (BJFC 024729); Tonkin, Cho-Ganh, sur branchettes de Flamboyant, 13 Mar 1914, L. Duport no. 14 (FH-HUH00940159, as *H. roseola*); sur “Dong” (bark of twigs), Janvier 1922, M. Duport no. 127 (FH-HUH00940157, as *H. roseol*a) and M. Duport no. 132 pp (FH-HUH01093669, as *H. tonkiniana*).

Notes—*Eichleriella discolor* is characterized by subceraceous-to-chalky or cretaceous basidiomes, small hyphal pegs arising from subiculum and composed of brownish yellow, sclerified hyphae and dikaryophyses, and 4-sterigmate basidia. The margin of specimens can be quite variable, and hyphal pegs are brittle, often breaking off to expose its dark brown interior. The size of the basidiospores is also variable and may reflect the number of sterigmata produced by the basidia. Its hyphal system was described as pseudodimitic [36], because the hyphae in the subiculum and hyphal peg are brownish yellow with slightly thickened walls. On close inspection, however, rare clamp connections were observed on these sclerified hyphae.

*Eichleriella discolor* is widely distributed from East and Southeast Asia to New Zealand and has been described several times. The synonymy presented above was reached after studying type specimens and mostly supports previous reports except that the synonymy of *H. roseola* with *K. discolor* is new. Bodman [16] considered *K. discolor* and *H. tonkiniana* to be synonyms but recognized *H. roseola* as a distinct species. Earlier, Roberts and Spooner [36] proposed the synonymy of *K. discolor* and *H. cheesmanii.*

**Eichleriella sinensis** (Teng) S.H. He & Nakasone, **comb. nov.** Figure 11C,D

MycoBank: MB847448

≡ *Heterochaete sinensis* Teng, Sinensia 7: 530, 1935.

= *Eichleriella aculeobasidiata* Hui Wang, Dong-Qiong Wang & C.L. Zhao, Kew Bull. 77: 326, 2022.

Fruiting body—Basidiomes widely effused, resupinate, first as small orbicular colonies, later confluent, adnate, moderately thick, (150–) 200–450 (–500) μm thick, firm, subceraceous to ceraceous. Hymenophore distinctly odontoid from hyphal pegs, (4–) 5–8 (–10) per mm, with smooth areas among pegs; pegs single, conical, terete, concolorous to brown, brittle, breaking off to reveal dark brown trama overall surface yellowish white (4A2), greyish yellow (4B3), pale orange (6A3), greyish orange (6B4), brownish orange (6C4), light brown [6D(4–5)] to dark brown (7F8), black in KOH then fading on drying; margins distinct, abrupt, detached, sometimes incurved, pale orange (5B3), light orange (5A4), greyish orange [5B(4-5)], often darker than hymenophore, with sterile edges up to 1 mm wide, even to scalloped, felty-woolly, rarely fimbriate.

Microscopic structures—Hyphal system dimitic with clamps on generative hyphae and thick-walled skeletal hyphae. Hyphal pegs arising from upper subiculum, 80–140 (–160) × (30–) 40–50 (–80) μm, a tight fascicle of non-agglutinated skeletal hyphae and coarse, colorless crystals, along sides and at apex skeletal hyphae differentiating into branched, thin-walled dikaryophyses. Subiculum up to 400 μm thick, composed of two distinct layers; (1) a non-agglutinated tissue, up to 200 μm thick, of dense, non-agglutinated, skeletal hyphae and degraded generative hyphae arranged parallel to substrate; (2) an interwoven tissue of non-agglutinated primarily skeletal hyphae and degraded subicular hyphae turning into hymenium, up to 250 μm thick, often with embedded, coarse colorless crystals; subicular hyphae mostly collapsed, 2–3 μm diam, clamped, walls colorless, thin; skeletal hyphae 2–4.5 μm diam, aseptate, sometimes with adventitious septa, rarely branched, walls pale yellow to brownish yellow, up to 2 μm thick, smooth. Subhymenium not observed. Hymenium up to 80 μm thick, a dense palisade of dikaryophyses enclosing cystidia and basidia. Dikaryophyses abundant, 30–60 × 3–4.5 μm, clamped at base, with short, knobby branches, walls colorless, thin, smooth. Cystidia numerous, embedded, cylindrical, clavate, or subfusiform, often stalked, apices rounded or subacute, 15–50 (–70) × 7.5–9 (–12.5) μm, clamped at base, walls colorless, thin, smooth, with homogenous contents. Basidia embedded, ovoid to ellipsoid, longitudinally septate, four-celled, 15–20 (–25) × 9–11 (–12.5) μm, with a small, basal clamp, 4-sterigmate, walls colorless, thin, smooth. Basidiospores scarce-to-numerous, cylindrical-to-narrowly cylindrical, sometimes ventrally depressed, (9.5–) 10.5–16 (–18) × (4.5–) 5.5–7 μm, L = 15 μm, W = 6.5 μm, Q = 2.2–2.4 (60/2), walls colorless, thin, smooth, IKI–, CB–, germination sometimes observed.

Distribution—Pantropical and subtropical: Australia, China, Indonesia, Japan, New Zealand, Philippines.

Specimens examined—Australia, Queensland, Cook, Innisfail, Joint Tropical Research Unit, on fallen, rotten twigs, 5 September 1972, N.E.M. Walters (MEL-231548, with *E. discolor*); China, Anhui Province, Chiu-hua-shan, (on small twigs), 18 September 1933, S.C.Teng 698 (paratype, BPI-719773); Fujian Province, Wuyishan County, Wuyishan Nature Reserve, Huanggangshan, on fallen angiosperm branch, 21 October 2005, Dai 7361 (BJFC 016640); Jian’ou County, Wanmulin Nature Reserve, on dead angiosperm branch, 19 August 2016, He 4547 (BJFC 023988); Guangxi Autonomous Region: Jinxiu County, Dayaoshan Nature Reserve, Shengtang Mountains, on rotten angiosperm trunk, 15 July 2017, He 4862 (BJFC 024381); Yinshan Park, on dead angiosperm branch, 16 July 2017, He 4900 (BJFC 024419); Guizhou Province, Suiyang County, Kuankuoshui Nature Reserve, on fallen angiosperm trunk, 26 November 2014, Dai 15044 (BJFC 018157); Libo County, Maolan Nature Reserve, on dead angiospermic branch, 14 June 2016, He 3761 (BJFC 022260, CFMR); Hainan Province, Baisha County, Yinggeling Nature Reserve, on dead angiosperm branch, 9 June 2016, He 3890 (BJFC 022392) and He 3892 (BJFC 022394); Henan Province, Xinyang County, Jigongshan Nature Reserve, on fallen Fagus branch, 27 October 2014, He 20141027-4 (BJFC 019266); Hubei Province, Wufeng County, on dead angiosperm branch, 14 August 2017, He 5025 (BJFC 024543); 15 August 2017, He 5057 (BJFC 024575); Hunan Province, Heng-shan, 22 September 1933, C.I. Shen 559 (paratype, BPI-719724); Dong’an County, Shunhangshan Nature Reserve, on dead angiospermic branch, 13 July 2015, He 2388 (BJFC 020842, CFMR); Liuyang County, Daweishan Forest Park, on dead angiosperm branch, 10 July 2015, He 2330 (BJFC 020784); Jilin Province: Jiaohe County, on dead angiosperm branch, 3 Sep. 2017, He 5145 (BJFC 024663). Jiangxi Province, Yifeng County, Guanshan Nature Reserve, on fallen angiosperm trunk, 9 August 2016, He 4185 (BJFC 023627); on dead angiosperm branch, 9 August 2016, He 4196 (BJFC 023638); Anyuan County, Sanbaishan Forest Park, on dead angiosperm branch, 15 August 2016, He 4425 (BJFC 023866); Sichuan Province, Ya’an, Bifengxia Forest Park, on bark of living Cunninghamia, 22 September 2012, He 20120922-4 (BJFC 014644); Qionglai County, Tiantaishan Forest Park, on dead angiosperm tree, 23 September 2012, He 20120923-18 (BJFC 014665); Taiwan, Nantou County, Jenai Township, Entrance of Southern Tungyenshan, alt. 1700 m, on dead angiosperm branch, 7 December 2016, He 4598 (BJFC 024040); Yunnan Province, Xichou County, Lotus pond, on fallen angiosperm branch, 25 July 2014, He 20140725-12 (BJFC 019206); Binchuan County, Jizhushan Forest Park, on dead angiosperm branch, 28 October 2017, He 5327 (BJFC 024845); Zhejiang Province, Tunglu, on dead twigs, 26 November 1933, S.Q. Deng 699 (paratype, BPI-719723; CUP-CH-000942); Indonesia, West Java, 1921, C. Hartley (BPI-719743, as *H. tenuicula*); Japan, no location data, K. Aoshima 10751 (NY-00461157, as *H. tenuicula*); New Zealand, Auckland, Mt. Albert Rd., on Tecoma sp., 6 November 1948, G.L.S. Chamberlain (PDD-24331, as *H. tenuicula*); Philippines, Quizon, Bataan, Lamae, on dead twigs, 30 November 1960, F.R. Vyenco (NY00461200, as *H. delicata*); Diliman, U.P. Campus, on bark, 7 August 1959, F.R. Vyenco (NY-00461151, as *H. delicata*); Los Banos, Luzon, Mt. Maquiling, on Cryptocarya dead branches, 30 July 1918, O.A. Reinking 7587 (BPI-719745, as *H. tenuicula*).

Notes—*Eichleriella sinensis* is similar to *E. tenuicula* except for the 4-sterigmate basidia and cylindrical basidiospores, average Q values 2.2–2.4, compared with allantoid basidiospores, average Q value 3.3 of the latter. The two taxa also overlap in distribution in Asia, Australia, New Zealand, and the Philippines but *E. tenuicula* has a pantropical distribution. Basidiospores were scarce in most of the specimens examined. Although the type specimen of *H. sinensis* was not examined, three paratypes were studied. 

In the phylogenetic tree of *Eichleriella* (Figure 3), *E. sinensis*, *E. tenuicula* and *E. delicata* are closely related but formed distinct lineages. The recently described species, *E. aculeobasidiata* Hui Wang, Dong Qiong Wang & C.L. Zhao, has 2-sterigmate basidia but nested within the *E. sinensis* lineage [28]. Close inspection of basidia of the type specimens of *E. aculeobasidiata* needs to be carried out, but at present we accept *E. aculeobasidiata* as a synonym of *E. sinensis* based on phylogenetic analyses and overall morphology. A sample from Japan identified as *Heterochaete delicata* (TUFC33717) in GenBank also nested within the *E. sinensis* lineage instead of the *E. delicata* lineage in our phylogenetic tree (Figure 3).

***Heteroradulum maolanense*** S.H. He, T. Nie & Yue Li, **sp. nov.** Figure 5E, Figure 12A and Figure 13

MycoBank: MB847449

Type—China, Guizhou Province, Libo County, Maolan Nature Reserve, on dead but still attached angiosperm branch, 11 July 2017, He 4773 (BJFC 024290, holotype, CFMR, isotype).

Etymology—refers to the type locality in Maolan Nature Reserve, Guizhou Province, southwestern China.

Fruiting body—Basidiomes annual, resupinate, effused, closely adnate, inseparable from substrate, coriaceous, first as small, scattered colonies, later confluent up to 10 cm long, 2.5 cm wide, up to 600 µm thick in section (hyphal pegs excluded). Hymenophore odontoid, irpicoid to subporoid from hyphal pegs, light orange [5A(4–5)], greyish orange [5B(4–6)] to brownish orange [5C(4–6)], not cracked; hyphal pegs densely arranged, single or fused, with blunt tips; margin abrupt, sterile, white (6A1) when juvenile, becoming concolorous with hymenophore with age.

Microscopic structures—Hyphal system monomitic-to-subdimitic, generative hyphae with clamp connections. Subiculum thick, pale yellow; hyphae colorless to pale yellow, slightly to distinctly thick-walled, rarely branched, septate, densely interwoven, smooth or encrusted with small granules, 2–3.5 µm in diam. Hyphal pegs originated from base of hymenium, pale yellow, up to 240 × 160 µm; hyphae similar to those of subiculum, agglutinated, vertically arranged, with tips heavily encrusted with crystals. Cystidia absent. Dikaryophyses numerous, colorless, thin-walled, encrusted, not branched or slightly branched at the tip. Basidia ovoid to subglobose, longitudinally septate, four-celled, without enucleate stalk, 18–25 × 10–15 µm. Basidiospores cylindrical with an apiculus, sometimes slightly curved, colorless, thin-walled, smooth, IKI–, CB–, containing one or two globules, capable of germinating by repetition, 12–15 × 6–7.5 µm, L = 16.5 µm, W = 7 µm, Q = 1.9–2 (*n* = 90/3).

Additional specimens examined—China, Guizhou Province, Libo County, Maolan Nature Reserve, on bark of dead angiosperm tree, 11 July 2017, He 4786 (BJFC 024303, CFMR); on dead but still attached angiosperm branch, 15 June 2016, He 3788 (BJFC 022287, CFMR).

Notes—*Heteroradulum maolanense* is characterized by its odontioid, irpicoid-to-subporoid hymenophore with single or fused hyphal pegs, encrusted hyphae and cylindrical basidiospores. The species is similar to *H. mussooriense* which differs in having wider ellipsoid-to-ovoid basidiospores (12–15 × 9–10 µm, [16]). *Heteroradulum australiense* L.W. Zhou, Q.Z. Li & S.L. Liu like *H. maolanense* has an odontoid hymenophore but differs in having both simple- and nodose-septate generative hyphae and longer basidiospores (15–20 × 5–7 µm, [20]). In the phylogenetic tree, the two samples of *H. maolanense* formed a distinct, strongly supported (Figure 4).

***Heteroradulum mussooriense*** (Bodman) S.H. He, T. Nie & Yue Li, **comb. nov.**

                                        Figure 12B

MycoBank: MB847450

≡ *Heterochaete mussooriensis* Bodman, Lloydia 15: 221, 1952.

= *Heteroradulum semis* Spirin & Malysheva, Fungal Biology 121: 712, 2017.

≡ *Grammatus semis* (Spirin & Malysheva) H.S. Yuan & Decock, MycoKeys 35: 35, 2018.

Specimens studied—China, Hubei Province, Shenlongjia Nature Reserve, on fallen angiosperm branch, 16 October 2016, Dai 17193 (BJFC 023291, CFMR); Yunnan Province, Binchuan County, Jizhushan Forest Park, on fallen angiosperm branch, 30 August 2015, He 2867 (BJFC 021301, CFMR); 26 November 2015, He 3168 (BJFC 021563, CFMR); 28 October 2017, He 5306 (BJFC 024824, CFMR) and He 5331 (BJFC 024849); Yongde County, Daxueshan Nature Reserve, on *Quercus* stump, 28 August 2015, He 2756 (BJFC 021194).

Notes—*Heterochaete mussooriensis* was described from Mussoorie, northwestern India, and then also reported in China [16,37]. The species is characterized by its subporoid hymenophore formed by numerous densely arranged spines, large subclavate basidia and broadly ellipsoid-to-ovoid basidiospores [16]. Malysheva and Spirin [15] described *Heteroradulum semis* from northeastern China and Japan. However, the protologue and our phylogenetic analyses show that *H. semis* is conspecific with *Heteroradulum mussooriensis*.

***Heteroradulum labyrinthinum*** (H.S. Yuan & Decock) L.W. Zhou, MycoKeys 86: 97, 2022.

                                        Figure 5F and Figure 12C,D

≡ *Grammatus labyrinthinus* H.S. Yuan & Decock, MycoKeys 35: 32, 2018.

Specimens examined—China, Guangxi Autonomous Region, Longzhou County, Nonggang Nature Reserve, on fallen angiosperm branch, 22 July 2012, He 20120722-1 (BJFC 014502, CFMR); Guizhou Province, Chishui County, Suoluo Nature Reserve, on fallen angiosperm branch, 7 July 2018, He 5431 (BJFC 026492) and He 5439 (BJFC026500).

Notes—Yuan et al. [30] established the genus, *Grammatus* H.S. Yuan & Decock, based on *G. labyrinthinus* and proposed the combination *G. semis*. The genus was later regarded as a synonym of *Heteroradulum* [20] and is also supported by our phylogenetic analyses (Figure 2). In the phylogenetic tree of *Heteroradulum* (Figure 4), three species, *H. labyrinthinum*, *H. mussooriense* and *H. maolanense*, form a strongly supported clade. Morphologically, the three species have similar and varied hymenophore that can be odontioid, irpicoid to subporoid because of the confluence the hyphal pegs at different growth stages (Figure 12). *Heteroradulum labyrinthinum* was originally described from tropical areas of Yunnan Province; herein, we report it from Guanxi Autonomous Region and Guizhou Province.

## 4. Discussion

In this study, we carried out a taxonomic and phylogenetic study of the Auriculariaceae and the genera *Eichleriella* and *Heteroradulum* with emphasis on corticioid samples from East and Southeast Asia. We resolved five new lineages represented by a new genus and five new species. In addition, four new combinations are proposed. Previous studies [4,15,21,22,23] and our results demonstrate that the Auriculariaceae is rich in corticioid fungi. However, additional studies are needed to understand the full scope of their diversity in subtropic and tropic regions. Some substrates, such as bamboo, appear to host unusual and undescribed taxa for further investigations. The taxonomy and phylogeny of the Auriculariaceae are far from resolved; however, many described species in *Exidiopsis*, *Heterochaete,* and *Exidia* have not been sequenced or studied critically by morphology. To create a robust family tree, future phylogenetic studies in the Auriculariaceace should include sequence data from multiple loci from a wide array of taxa, including taxa described from the last century, as well as taxa representing other regions of the world.


**key to corticioid and stereoid genera in Auriculariaceae**


1. Hymenophore strictly smooth21. Hymenophore smooth, odontoid, irpicoid, or subporoid42. Hyphal system dimitic
*Amphistereum*
2. Hyphal system monomitic33. Basidiospores cylindrical or ellipsoid
*Heterocorticium*
3. Basidiospores allantoid, distinctly curved
*Sclerotrema*
4. Basidiomes leathery54. Basidiomes gelatinous, waxy or crustaceous75. Basidiomes resupinate, smooth or with small spines
*Alloexidiopsis*
5. Basidiomes effused-reflexed, smooth or with distinct spines66. With encrusted hyphal ends
*Heterordulum*
6. Without encrusted hyphal ends
*Eichleriella*
7. Spines fertile
*Proterochaete*
7. Spines (hyphal pegs) sterile88. Hyphal pegs heavily covered with crystals98. Hyphal pegs smooth109. Hyphal pegs covered by angular crystals
*Crystallodon*
9. Hyphal pegs covered with metuloid cystidia
*Metulochaete*
10. Basidiospores subglobose, > 9 μm wide
*Hirneolina*
10. Basidiospores cylindrical, < 9 μm wide1111. Hymenophore grayish to brownish
*Adustochaete*
11. Hymenophore pale-colored
*Exidiopsis*


## Figures and Tables

**Figure 1 jof-09-00318-f001:**
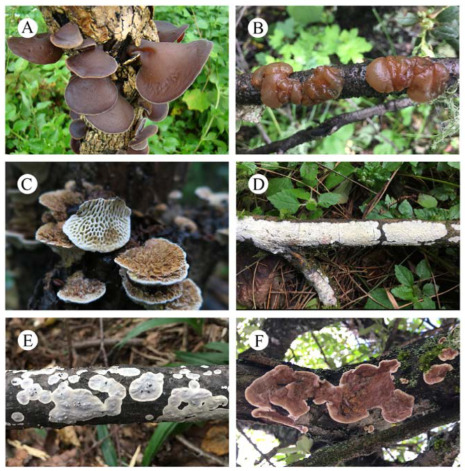
Representative taxa of Auriculariaceae. (**A**). *Auricularia cornea* Ehrenb.; (**B**). *Exidia* sp.; (**C**). *Elmerina* sp.; (**D**). *Alloexidiopsis calcea* (Pers.) L.W. Zhou and S.L. Liu; (**E**). *Eichleriella crocata* (Pat.) Spirin and Malysheva; (**F**). *Heteroradulum kmetii*.

**Figure 2 jof-09-00318-f002:**
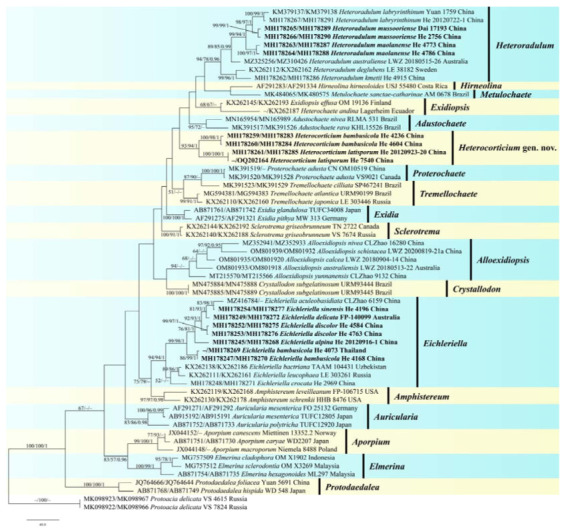
Phylogenetic tree of MP analysis from the concatenated ITS-28S sequences of Auriculariaceae taxa. Branches are labelled with parsimony bootstrap values (≥ 50%, first), likelihood bootstrap values (≥ 50%, second), and Bayesian posterior probabilities (≥ 0.95, third). New species and new combinations are set in bold.

**Figure 3 jof-09-00318-f003:**
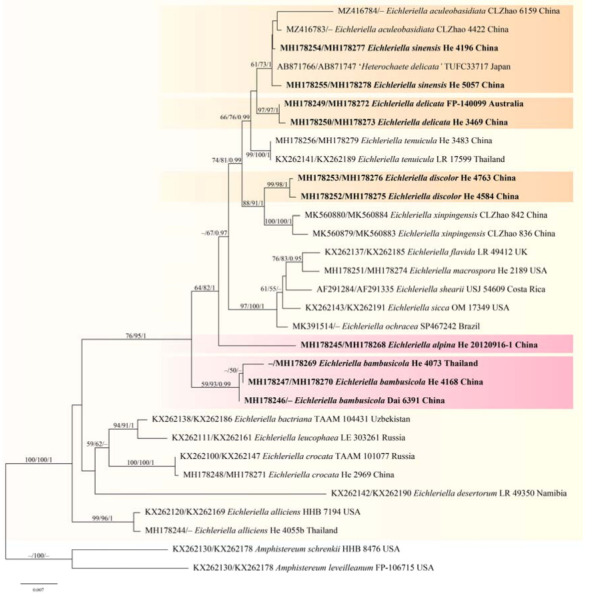
Phylogenetic tree of ML analysis from the concatenated ITS-28S sequences of *Eichleriella* species. Branches are labelled with parsimony bootstrap values (≥ 50%, first), likelihood bootstrap values (≥ 50%, second), and Bayesian posterior probabilities (≥ 0.95, third). New species (pink) and new combinations (orange) are highlighted and set in bold.

**Figure 4 jof-09-00318-f004:**
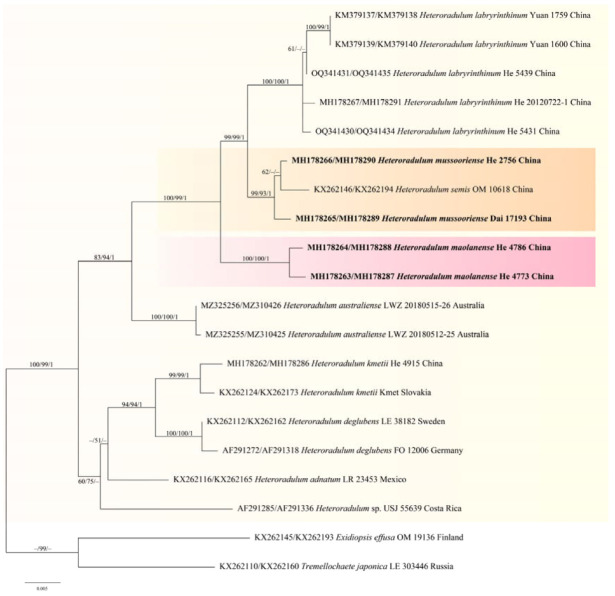
Phylogenetic tree of ML analysis of the concatenated ITS-28S sequences of *Heteroradulum* species. Branches are labelled with parsimony bootstrap values (≥ 50%, first), likelihood bootstrap values (≥ 50%, second), and Bayesian posterior probabilities (≥ 0.95, third). New species (pink) and new combinations (orange) are highlighted and set in bold.

**Figure 5 jof-09-00318-f005:**
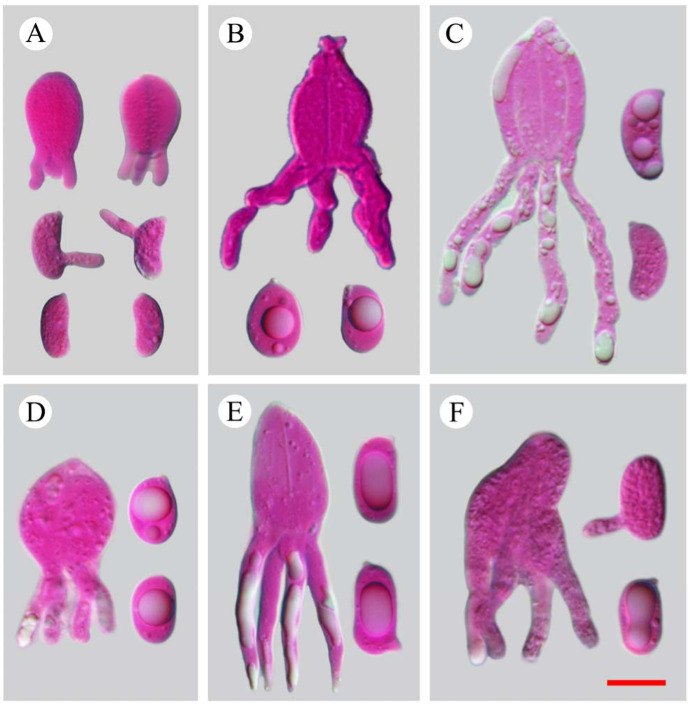
Micrographs of basidia and basidiospores (in phloxine and KOH). Scale bar = 10 µm. (**A**). *Heterocorticium bambusicola* (holotype, He 4236, BJFC 023678); (**B**). *Heterocorticium latisporum* (holotype, He 20120923-20, BJFC 014667); (**C**). *Eichleriella alpina* (holotype, He 20120916-1, BJFC 014581); (**D**). *Eichleriella bambusicola* (holotype, He 4073, BJFC 023514); (**E**). *Heteroradulum maolanense* (holotype, He 4773, BJFC 024290); (**F**). *Heteroradulum labyrinthinum* (He 20120722-1, BJFC 014502).

**Figure 6 jof-09-00318-f006:**
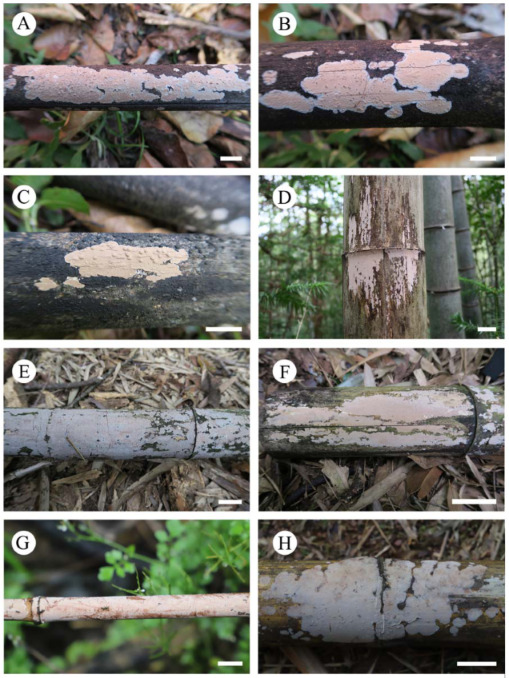
Basidiomes of *Heterocorticium bambusicola*. Scale bars: (**A–H**) = 1 cm. (**A**). He 3341 (BJFC 021736); (**B**). He 3344 (BJFC 021739); (**C**). He 3359 (BJFC 021754); (**D**). He 4234 (BJFC023676); (**E**). He 4604 (BJFC 024046); (**F**). He 4607 (BJFC 024049); (**G**). He 6293 (BJFC 033237); (**H**). He 6435 (BJFC 033379).

**Figure 7 jof-09-00318-f007:**
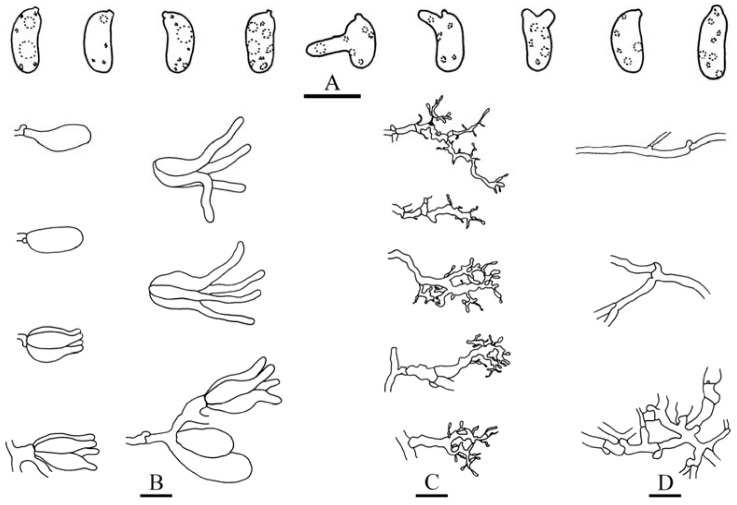
*Heterocorticium bambusicola* (from the holotype, He 4236, BJFC 023678). Scale bars: (**A–D**) = 10 µm. (**A**). Basidiospores; (**B**). Basidia; (**C**). Dikaryophyses; (**D**). Generative hyphae.

**Figure 8 jof-09-00318-f008:**
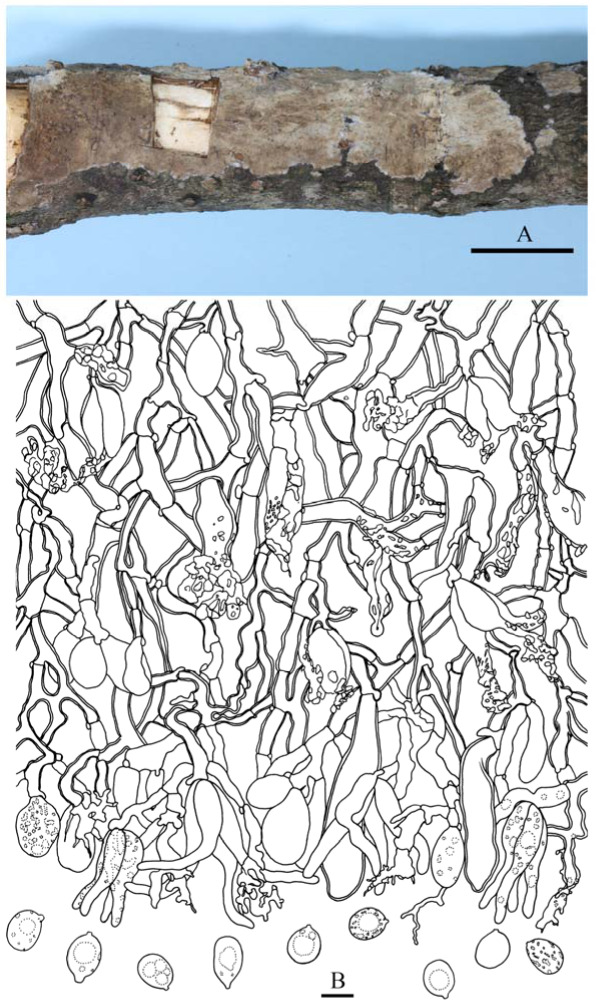
*Heterocorticium latisporum* (from the holotype, He 20120923-20, BJFC 014667). Scale bars: (**A**) = 1 cm; (**B**) = 10 µm. (**A**). Basidiomes; (**B**). A cross section of the basidiome.

**Figure 9 jof-09-00318-f009:**
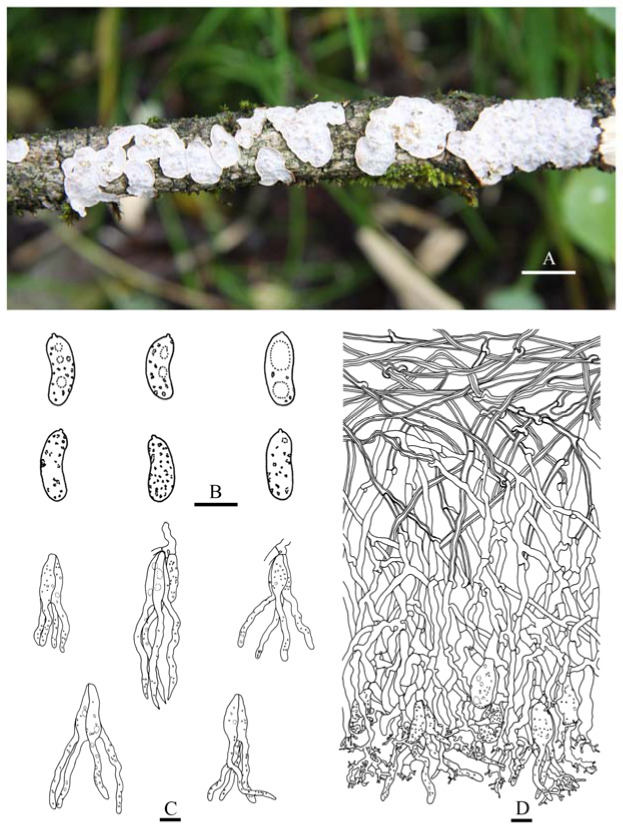
*Eichleriella alpina* (from the holotype, He 20120916-1, BJFC 014581). Scale bars: (**A**) = 1 cm; (**B**–**D**) = 10 µm. (**A**). Basidiomes; (**B**). Basidiospores; (**C**). Basidia; (**D**). A cross section of the basidiome.

**Figure 10 jof-09-00318-f010:**
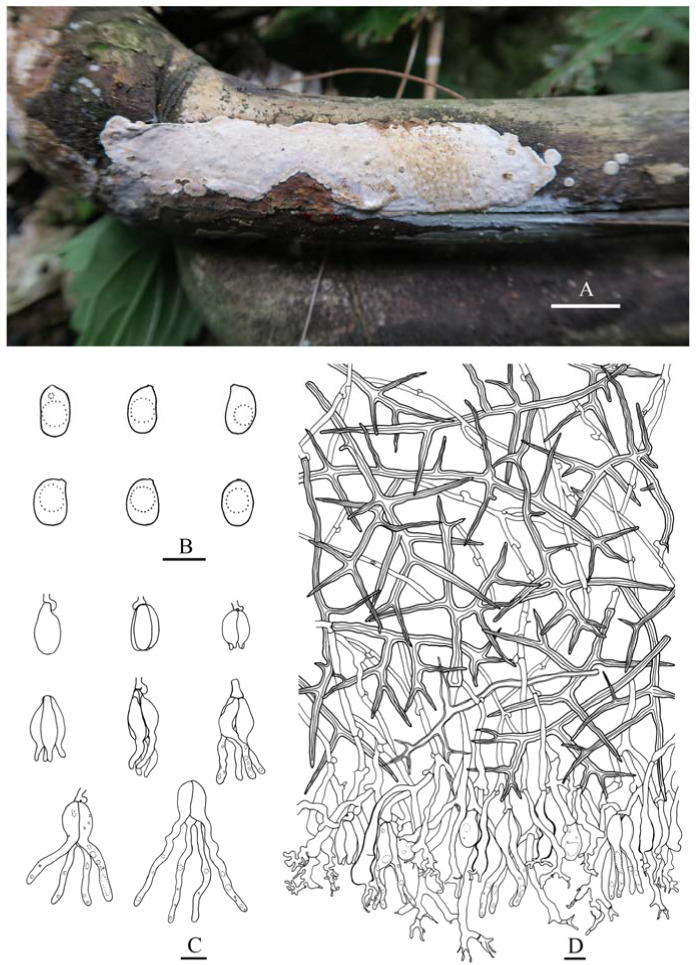
*Eichleriella bambusicola* (from the holotype, He 4073, BJFC 023514). Scale bars: (**A**) = 1 cm; (**B–D**) = 10 µm. (**A**). Basidiomes; (**B**). Basidiospores; (**C**). Basidia; (**D**). A cross section of the basidiome.

**Figure 11 jof-09-00318-f011:**
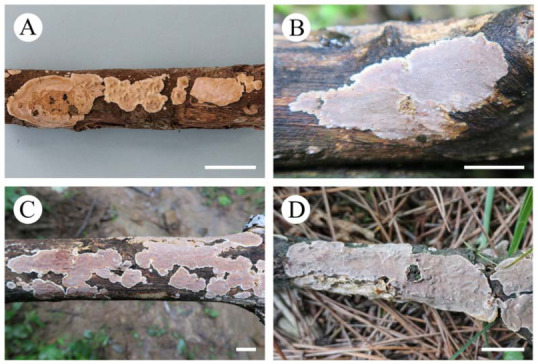
Basidiomes of *Eichleriella* species. Scale bars: (**A–D**) = 1 cm. (**A**). *E. delicata* (He 3469, BJFC 021866); (**B**). *E. discolor* (He 4763, BJFC 024281); (**C**). *E. sinensis* (He 4185, BJFC 023627); (**D**). *E. sinensis* (He 5392, BJFC 026453).

**Figure 12 jof-09-00318-f012:**
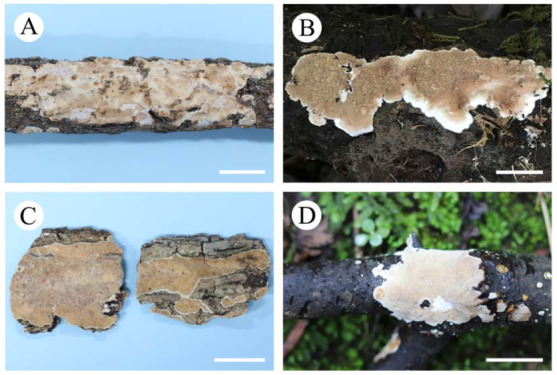
Basidiomes of *Heteroradulum* species. Scale bars: (**A–D**) = 1 cm. (**A**). *H. maolanense* (holotype, He 4773, BJFC 024290); (**B**). *H. mussooriense* (He 5331, BJFC 024849); (**C**). *H. labyrinthinum* (He 20120722-1, BJFC 014502); (**D**). *H. labyrinthinum* (He 5431, BJFC 026492).

**Figure 13 jof-09-00318-f013:**
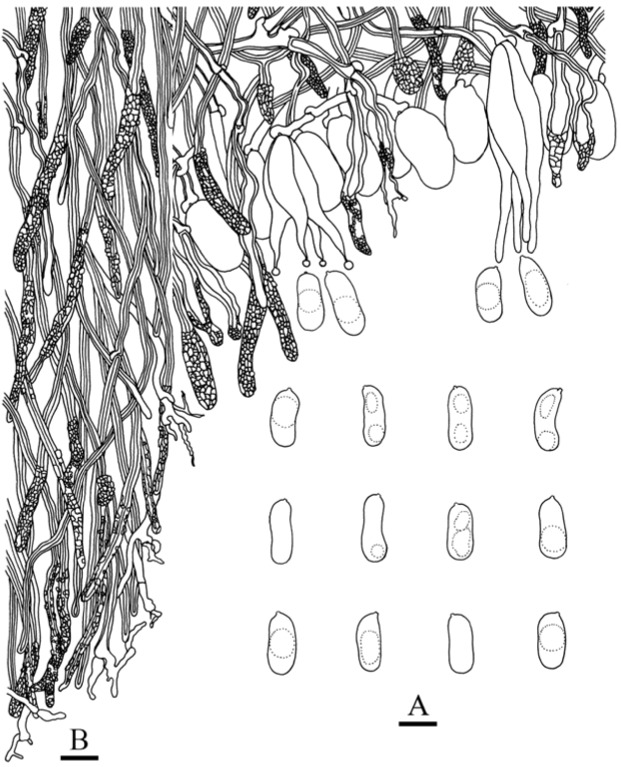
*Heteroradulum maolanense* (from the holotype, He 4773, BJFC 024290). Scale bars: (**A**,**B**) = 10 µm. (**A**). Basidiospores; (**B**). A cross section of the basidiome.

**Table 1 jof-09-00318-t001:** Species and sequences used in the phylogenetic analyses. New taxa are set in bold with type specimens indicated with an asterisk (*).

Species	Specimen No.	Locality	GenBank Accession No.		Reference
			ITS	28S	
*Adustochaete nivea*	RLMA 531	Brazil	MN165954	MN165989	[25]
*Adustochaete rava*	KHL15526	Brazil	MK391517	MK391526	[22]
*Alloexidiopsis australiensis*	LWZ 20180513-22	Australia	OM801933	OM801918	[4]
*Alloexidiopsis calcea*	LWZ 20180904-14	China	OM801935	OM801920	[4]
*Alloexidiopsis nivea*	CLZhao 16280	China	MZ352941	MZ352933	[26]
*Alloexidiopsis schistacea*	LWZ 20200819-21a	China	OM801939	OM801932	[4]
*Alloexidiopsis yunnanensis*	CLZhao 9132	China	MT215570	MT215566	[27]
*Amphistereum leveilleanum*	FP-106715	USA	KX262119	KX262168	[15]
*Amphistereum schrenkii*	HHB 8476	USA	KX262130	KX262178	[15]
*Aporpium canescens*	Miettinen 13352.2	Norway	JX044152	—	[12]
*Aporpium caryae*	WD2207	Japan	AB871751	AB871730	[14]
*Aporpium macroporum*	Niemela 8488	Poland	JX044148	—	[12]
*Auricularia mesenterica*	FO 25132	Germany	AF291271	AF291292	[1]
*Auricularia mesenterica*	TUFC12805	Japan	AB915192	AB915191	[14]
*Auricularia polytricha*	TUFC12920	Japan	AB871752	AB871733	[14]
*Crystallodon subgelatinosum*	URM93444	Brazil	MN475884	MN475888	[23]
*Crystallodon subgelatinosum*	URM93445	Brazil	MN475885	MN475889	[23]
*Eichleriella aculeobasidiata*	CLZhao 4422	China	MZ416783	—	[28]
*Eichleriella aculeobasidiata*	CLZhao 6159	China	MZ416784	—	[28]
*Eichleriella alliciens*	He 4055b	Thailand	MH178244	—	Present Study
*Eichleriella alliciens*	HHB 7194	USA	KX262120	KX262169	[15]
** *Eichleriella alpina* **	**He 20120916-1 ***	**China**	**MH178245**	**MH178268**	**Present Study**
*Eichleriella bactriana*	TAAM 104431	Uzbekistan	KX262138	KX262186	[15]
** *Eichleriella bambusicola* **	**Dai 6391**	**China**	**MH178246**	**—**	**Present Study**
** *Eichleriella bambusicola* **	**He 4073 ***	**Thailand**	**—**	**MH178269**	**Present Study**
** *Eichleriella bambusicola* **	**He 4168**	**China**	**MH178247**	**MH178270**	**Present Study**
*Eichleriella crocata*	He 2969	China	MH178248	MH178271	Present Study
*Eichleriella crocata*	TAAM 101077	Russia	KX262100	KX262147	[15]
** *Eichleriella delicata* **	**FP-140099**	**Australia**	**MH178249**	**MH178272**	**Present Study**
** *Eichleriella delicata* **	**He 3469**	**China**	**MH178250**	**MH178273**	**Present Study**
*Eichleriella desertorum*	LR 49350	Namibia	KX262142	KX262190	[15]
** *Eichleriella discolor* **	**He 4584**	**China**	**MH178252**	**MH178275**	**Present Study**
** *Eichleriella discolor* **	**He 4763**	**China**	**MH178253**	**MH178276**	**Present Study**
*Eichleriella flavida*	LR 49412	UK	KX262137	KX262185	[15]
*Eichleriella leucophaea*	LE 303261	Russia	KX262111	KX262161	[15]
*Eichleriella macrospora*	He 2189	USA	MH178251	MH178274	Present Study
*Eichleriella ochracea*	SP467242	Brazil	MK391514	—	[22]
*Eichleriella shearii*	USJ 54609	Costa Rica	AF291284	AF291335	[1]
*Eichleriella sicca*	OM 17349	USA	KX262143	KX262191	[15]
** *Eichleriella sinensis* **	**He 4196**	**China**	**MH178254**	**MH178277**	**Present Study**
** *Eichleriella sinensis* **	**He 5057**	**China**	**MH178255**	**MH178278**	**Present Study**
*Eichleriella tenuicula*	He 3483	China	MH178256	MH178279	Present Study
*Eichleriella tenuicula*	LR 17599	Thailand	KX262141	KX262189	[15]
*Eichleriella xinpingensis*	CLZhao 836	China	MK560879	MK560883	[29]
*Eichleriella xinpingensis*	CLZhao 842	China	MK560880	MK560884	[29]
*Elmerina cladophora*	OM X1902	Indonesia	MG757509	MG757509	[21]
*Elmerina hexagonoides*	ML297	Malaysia	AB871754	AB871735	[14]
*Elmerina sclerodontia*	OM X3269	Malaysia	MG757512	MG757512	[21]
*Exidia glandulosa*	TUFC34008	Japan	AB871761	AB871742	[14]
*Exidia pithya*	MW 313	Germany	AF291275	AF291321	[1]
*Exidiopsis effusa*	OM 19136	Finland	KX262145	KX262193	[15]
*Heterochaete andina*	Lagerheim	Ecuador	—	KX262187	[15]
*Heterochaete delicata*	TUFC33717	Japan	AB871766	AB871747	[14]
** *Heterocorticium bambusicola* **	**He 4236 ***	**China**	**MH178259**	**MH178283**	**Present Study**
** *Heterocorticium bambusicola* **	**He 4604**	**China**	**MH178260**	**MH178284**	**Present Study**
** *Heterocorticium latisporum* **	**He 20120923-20 ***	**China**	**MH178261**	**MH178285**	**Present Study**
** *Heterocorticium latisporum* **	**He 7540**	**China**	**—**	**OQ202164**	**Present Study**
*Heteroradulum adnatum*	LR 23453	Mexico	KX262116	KX262165	[15]
*Heteroradulum australiense*	LWZ 20180512-25	Australia	MZ325255	MZ310425	[20]
*Heteroradulum australiense*	LWZ 20180515-26	Australia	MZ325256	MZ310426	[20]
*Heteroradulum deglubens*	FO 12006	Germany	AF291272	AF291318	[1]
*Heteroradulum deglubens*	LE 38182	Sweden	KX262112	KX262162	[15]
*Heteroradulum labyrinthinum*	Yuan 1600	China	KM379139	KM379140	[30]
*Heteroradulum labyrinthinum*	Yuan 1759	China	KM379137	KM379138	[30]
*Heteroradulum labyrinthinum*	He 20120722-1	China	MH178267	MH178291	Present Study
*Heteroradulum labyrinthinum*	He 5431	China	OQ341430	OQ341434	Present Study
*Heteroradulum labyrinthinum*	He 5439	China	OQ341431	OQ341435	Present Study
*Heteroradulum kmetii*	He 4915	China	MH178262	MH178286	Present Study
*Heteroradulum kmetii*	Kmet	Slovakia	KX262124	KX262173	[15]
** *Heteroradulum maolanense* **	**He 4773 ***	**China**	**MH178263**	**MH178287**	**Present Study**
** *Heteroradulum maolanense* **	**He 4786**	**China**	**MH178264**	**MH178288**	**Present Study**
** *Heteroradulum mussooriense* **	**Dai 17193**	**China**	**MH178265**	**MH178289**	**Present Study**
** *Heteroradulum mussooriense* **	**He 2756**	**China**	**MH178266**	**MH178290**	**Present Study**
*Heteroradulum semis*	OM 10618	China	KX262146	KX262194	[15]
*Heteroradulum* sp.	USJ 55639	Costa Rica	AF291285	AF291336	[1]
*Hirneolina hirneoloides*	USJ 55480	Costa Rica	AF291283	AF291334	[1]
*Metulochaete sanctae-catharinae*	AM 0678	Brazil	MK484065	MK480575	[21]
*Proterochaete adusta*	CN OM10519	China	MK391519	—	[22]
*Proterochaete adusta*	VS9021	Canada	MK391520	MK391528	[22]
*Protoacia delicata*	VS 4615	Russia	MK098923	MK098967	[31]
*Protoacia delicata*	VS 7824	Russia	MK098922	MK098966	[31]
*Protodaedalea foliacea*	Yuan 5691	China	JQ764666	JQ764644	[13]
*Protodaedalea hispida*	WD 548	Japan	AB871768	AB871749	[14]
*Sclerotrema griseobrunneum*	TN 2722	Canada	KX262144	KX262192	[15]
*Sclerotrema griseobrunneum*	VS 7674	Russia	KX262140	KX262188	[15]
*Tremellochaete atlantica*	URM90199	Brazil	MG594381	MG594383	[32]
*Tremellochaete cilliata*	SP467241	Brazil	MK391523	MK391529	[22]
*Tremellochaete japonica*	LE 303446	Russia	KX262110	KX262160	[15]

## Data Availability

Not applicable.

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
