# Peer review of "Taxonomy and Phylogeny of Corticioid Fungi in Auriculariaceae (Auriculariales, Basidiomycota): A New Genus, Five New Species and Four New Combinations"

_jof, 2023, doi:10.3390/jof9030318_

Round 1
Reviewer 1 Report
Introduction - this part needs attention to the language; please ask a native English speaker to correct this part.
Please check carefully for each verb whether you have to use the plural or the singular form.
Indicate under material and methods where collections were made.
Elaborate on the choice of sequences for comparison. Which sequences are included determines the outcome of the analyses.
I also would like to see the rationale for using ITS and LSU, and not any protein coding gene for the phylogenetic analyses.
Heteroradulum is a word of neutral gender, the epithets should reflect that: Heteroradulum maolanense, H. mussouriense, and H. labryrinthinum, Please change the ending in all places (including phylogenetic trees and legends of figures).
in reference 21 and 37 use capital letters for the first letter of the scientific names.
Author Response
Q1: Introduction - this part needs attention to the language; please ask a native English speaker to correct this part.
R1: A native English speaker revised the language of the manuscript. We made many corrections on the language in the resubmitted manuscript.
Q2: Please check carefully for each verb whether you have to use the plural or the singular form.
R2: Yes, we did and corrected in many places.
Q3: Indicate under material and methods where collections were made.
R3: The exact locations were listed for each specimen. Also, we said in MS that the specimens studied in the paper were mostly from East and Southeast Asia.
Q4: Elaborate on the choice of sequences for comparison. Which sequences are included determines the outcome of the analyses. I also would like to see the rationale for using ITS and LSU, and not any protein coding gene for the phylogenetic analyses.
R4: For the choice of sequences, we selected those of the type species of each genus, and for each species, we select sequences of its holotype if available.
At present, there are many ITS and LSU sequences of Auriculariales in GenBank, but few of protein coding gene. So, it is possible to do the phylogenetic analyses of the whole family or order by using ITS and LSU sequences.
Q5: Heteroradulum is a word of neutral gender, the epithets should reflect that: Heteroradulum maolanense, H. mussouriense, and H. labryrinthinum, Please change the ending in all places (including phylogenetic trees and legends of figures).
R5: We corrected the names in the whole text and all figures.
Q6: In reference 21 and 37 use capital letters for the first letter of the scientific names.
R6: Yes, we corrected.
Reviewer 2 Report
The manuscript is really interesting and I think it is a great contribution. Only a few really minor corrections are needed: Heteroradulum is a neuter name, thus H. maolanensis and H. mussooriensis are wrongly declined and should be changed to H. maolanense and H. mussooriense respectively. Please correct this.
Heteroradulum labyrinthinum is correctly written along the manuscript, but misspelled as H. labyrinthinus in the table as well as in figure 5F and in the tree. Please correct this too.
When the authors indicate that H. latisporum grows angiosperm wood instead of bamboo, I would point out that bamboo is also an angiosperm. I would change de wording just to be more precise. this is just a suggestion.
Author Response
Q1: Heteroradulum is a neuter name, thus H. maolanensis and H. mussooriensis are wrongly declined and should be changed to H. maolanense and H. mussooriense respectively. Please correct this.
R1: We corrected the names in the whole text and all figures.
Q2: Heteroradulum labyrinthinum is correctly written along the manuscript, but misspelled as H. labyrinthinus in the table as well as in figure 5F and in the tree. Please correct this too.
R2: We have corrected all the misspelling about “H. labyrinthinus”.
Q3: When the authors indicate that H. latisporum grows angiosperm wood instead of bamboo, I would point out that bamboo is also an angiosperm. I would change de wording just to be more precise. this is just a suggestion.
R3: We deleted this point of difference in the revised MS.
Reviewer 3 Report
Measurements in µm should be given in 0.5 µm steps. So measurements like 1.7, 3.2, 4,4 etc. should be 2.0, 3.0 and 4.5. The reason is because it is not possible to measure beyond 0.2 µm in a light microscope theoretically. Practically only 0.5 µm.
While treating different genera in Auriculariaceae it would be of big help to have a diagnostic identification key in the manuscript. This would help the reader to understand the phylogenetic trees better.
Author Response
Q1: Measurements in µm should be given in 0.5 µm steps. So measurements like 1.7, 3.2, 4,4 etc. should be 2.0, 3.0 and 4.5. The reason is because it is not possible to measure beyond 0.2 µm in a light microscope theoretically. Practically only 0.5 µm.
R1: We have changed all the measurements to 0.5 or integer numbers.
Q2: While treating different genera in Auriculariaceae it would be of big help to have a diagnostic identification key in the manuscript. This would help the reader to understand the phylogenetic trees better.
R2: We added an identification key of corticioid and stereoid genera in Auriculariaceae.
Reviewer 4 Report
This manuscript presents a very nice systematic study within Auriculariaceae. Besides some issues with the English mostly in the introduction, and some areas of improvement for the phylogenetic inference, it is a well-presented study. The two most important issue is the omission of single-locus trees in the supplement, and the lack of geographic information about where the samples were collected.
Some of your data sets do not have complete coverage at a single locus - because ITS has the greater resolution, you should really omit samples that are only represented by a 28S sequence for the concatenated analyses (ie, Eichleriella bambusicolaHe 4073*, Heterocorticium latisporumHe 7540). ITS-only is ok.
It is important to include single-locus trees from all analyses (supplementally), as there are likely to be some topological differences, despite the two loci being so close. And, this would allow you to show the 28S-only samples that you should omit from the combined analysis. Although MAFFT g-INS-i is a good alignment method, for rDNA, MAFFT q-INS-i or x-INS-i is recommended.
The number of taxa is small enough that for the ML and BI inference, the performance would likely be improved by combining all three data sets into a single analysis. The figures could still be arranged as they are, with Eichleriella and Heteroradulum collapsed in the family-wide tree.
On the taxon-sampling note, it is unfortunate that there is only a single species sampled for both Heterochaete and Exidiopsis, as your tree cannot verify their monophyly. I'm not sure if this is because of a lack of accessions or if these are monotypic genera, but if there are more available at least one species of each genus should be added. The same is true for Hirneolina and Metulochaete.
Partitioning of the data for the ML and BI inference (and use of a ML program with access to more models, such as IQTREE-2) would likely improve the topology and support for the deeper branches of your tree. Furthermore, adding phylogenetically informative gaps (eg with FastGap) would also add more characters for the ML and BI analyses.
However, your clades are well-resolved and supported, so the improvements to the inference I suggest are not likely to change any of the systematic outcomes of the manuscript. The single-locus trees are the major issue I see.
pg 1
13 most of the species in the Auriculariales
32 well supported -> best-supported
34 were nested wtihin -> are placed in, are members of
34 has -> have
36 later -> latter
37 hymenophores
38 closely related in phylogeny -> phylogenetically close
38 similar to each other in morphology -> morphologically similar
39 omit "The"
41 omit "the"
56 has -> have
62 distinguishes -> distinction
66 results -> result
76 this section needs to include something about where the samples were collected, or point to a list of isolates that includes geographic data
110-118 this section needs to refer to the tables (in-manuscript or supplemental) where we can find the accessions used for the phylogeny
121-132 was there any partitioning for the ML or BI inference?
122 (TBR) is not used anywhere else in the manuscript - this abbreviation is unnecessary
128 How many "runs" were employed in the MrBayes analysis? Was there partitioning?
Author Response
Q1: The two most important issue is the omission of single-locus trees in the supplement, and the lack of geographic information about where the samples were collected.
R1: 1. For the single-locus tree: We performed additional phylogenetic analyses based on the ITS sequence. The results are consistent with the ITS-LSU analyses. please see the attached pdf file of the ITS tree. 2. For the geographic information of samples: The exact locations were listed for each specimen. Also, we said in MS that the specimens studied in the paper were mostly from East and Southeast Asia.
Q2: 13 most of the species in the Auriculariales
32 well supported -> best-supported
34 were nested wtihin -> are placed in, are members of
34 has -> have
36 later -> latter
37 hymenophores
38 closely related in phylogeny -> phylogenetically close
38 similar to each other in morphology -> morphologically similar
39 omit "The"
41 omit "the"
56 has -> have
62 distinguishes -> distinction
66 results -> result
R2: Most of the corrections were accepted, for few corrections, the whole sentences were rewritten or deleted in the revised MS.
Q3: 110-118 this section needs to refer to the tables (in-manuscript or supplemental) where we can find the accessions used for the phylogeny
R3: We reorganized the materials and methods section. However, we referred Table 1 in this section.
Q4: 122 (TBR) is not used anywhere else in the manuscript - this abbreviation is unnecessary
R4: We reorganized the materials and methods section, and the abbreviation of TBR was deleted.
Q5: 128 How many "runs" were employed in the MrBayes analysis? Was there partitioning?
R5: 131-134 We have mentioned the “runs” in our manuscript: Four Markov chains were run for 1000000 generations for Eichleriella and Heteroradulum datasets separately, 4000000 generations for the Auriculariaceae dataset until the split deviation frequency value was lower than 0.01.
